# The Impact of Health Literacy on Knowledge and Attitudes towards Preventive Strategies against COVID-19: A Cross-Sectional Study

**DOI:** 10.3390/ijerph18105421

**Published:** 2021-05-19

**Authors:** Maria João Silva, Paulo Santos

**Affiliations:** 1Department of Medicine of Community, Information and Health Decision Sciences (MEDCIDS), Faculty of Medicine, University of Porto, 4200-319 Porto, Portugal; up201403571@edu.med.up.pt; 2Center for Health Technology and Services Research (CINTESIS), Faculty of Medicine, University of Porto, 4200-319 Porto, Portugal

**Keywords:** health literacy, COVID-19, health behavior, attitudes, health knowledge, practice

## Abstract

The coronavirus disease 2019 (COVID-19) pandemic introduced a set of mitigation measures based on personal behavior and attitudes. In the absence of vaccination or specific treatment, it became essential to comply with these measures to reduce infection transmission. Health literacy is the basis for changing behaviors. AIM: To characterize the impact of literacy on knowledge and attitudes towards preventive strategies against COVID-19. METHODS: This cross-sectional study involved an online questionnaire applied to students of the University of Porto, Portugal, containing questions about knowledge and attitudes towards COVID-19 based on European guidelines. Health literacy was assessed through the Newest Vital Sign questionnaire. Logistic regression estimated the relationship between health literacy and both knowledge and attitudes. RESULTS: We included 871 participants (76.3% female), with a median age of 22 years old. We found adequate literacy in 92% of our sample, irrespective of gender and age. In the global analysis, 78.6% of the participants had adequate knowledge, and 90.4% had adequate attitudes. We found that better literacy was significantly associated with attitudes towards COVID-19, but not with better knowledge. In a model adjusted for gender, age, and previous education in the health field, female gender and previous education in the health field were associated with better knowledge and attitudes. CONCLUSION: Better health literacy is associated with better attitudes towards preventive strategies against COVID-19. We should invest in ways to improve health literacy, so we can improve people’s attitudes and consequently reduce coronavirus’ transmission.

## 1. Introduction

Early in 2020, the world experienced an unexpected health crisis. Suddenly, our lives changed dramatically as the new coronavirus, SARS-Cov-2, widely spread, leading to the declaration of the pandemic by the World Health Organization [1]. The onset of a new infectious threat, in a time where infectious diseases were no longer a public concern, raised old fears. Current technological capacity was not enough to protect people from disease and even death. The total impact extended far from health issues to the social, economic and political levels.

The fast development of coronavirus disease called for people to acquire health information and to apply it, adapting their behavior at a fast pace [2]. Since vaccination is still proceeding at a slow rate, and in the absence of a specific treatment, adopting non-pharmacological public health interventions became really important to combat COVID-19 by slowing down the virus transmission [3,4,5].

In this sense, literacy has become more relevant than ever. Ratzan, in 2000, settled the concept of literacy as the capacity to deal with basic health information by obtaining it, processing and understanding it, along with the knowledge of services needed to make appropriate health options [6]. Over the past years, literacy has assumed an important role in health communication, leaving the closed aspect of simple knowledge to a more functional perception of the activation of individuals for better behaviors, activating them and not only teaching them [7]. More than what each person knows, it matters what each person does. This higher complexity led to a new definition proposed by Berkman separating health literacy from intelligence, adding importance to oral communication skills as a critical component of health literacy, and eliminating the term “basics” [8].

Health literacy is not merely a function of basic skills. It includes the capacity to communicate about health issues, depending on individual and systemic factors, such as beliefs, culture, education, and health services organization, framed by healthcare needs and context demands [9]. Literacy is a relevant determinant for health, empowering individuals to participate and take action in their own healthcare [1], with an impact on health outcomes. Furthermore, health literacy improves health and well-being, addresses health inequalities, and builds individual and community resilience [1]. Moreover, it allows individuals to make better health decisions and have a stronger commitment and higher levels of efficiency [10]. On the other hand, low literacy is associated with higher mortality [11], higher morbidity [12], higher rates of depression [13], and lower adherence to medications [14,15], and commonly is correlated with lack of education, poverty, unemployment, and low socioeconomic status. Notwithstanding, even those with preeminent education and income can have low health literacy when experiencing something for the first time [1].

Since the beginning of the COVID-19 pandemic, a great amount of daily new data were delivered to the public from numerous official and non-official sources [16]. Comments and explanations of many experts and opinion makers invaded the news, sometimes in a contradictory way. The oversaturation of information made it hard to distinguish between correct and wrong information, allowing for the introduction of misconceptions and wrong beliefs, many times under the cover of almost scientific speech. Furthermore, the way people access information has changed over the years. Information sources are crucial to frame how to reach people. For the younger population, internet has substituted the classical media as the preferred source of mass information, with easily accessible and comprehensive content. However, internet is associated with lower literacy levels [17], because of the amount of fake information that is disseminated without technical review and appraisal [18]. Recent studies on digital health literacy about COVID-19 show that, although students generally achieve high levels, they are unable to make judgments about the reliability of online health information [19,20,21]. Without true information there is no real freedom to decide. Most of the time, this lack of autonomy results in low adherence and even rejection of preventive health measures [22]. Consequently, many people have adopted wrong behaviors against COVID-19.

Health literacy, both in knowledge as in behavior, is the key for an effective preventive medicine [10] and the way to a greater justice and equity in society as a long-term measure [23].

In this COVID-19 pandemic, Gautam et al. showed that health literacy is a significant predictor for awareness and preventive behaviors among chronic disease patients [24]. In the younger population, COVID-19 has shown low impact both in incidence and in the clinical severity. In this group, the relation between literacy and preventive COVID-19 behaviors has not yet been established, nor have its determinants, making hard to intervene preventively. The aim of this study is to characterize the impact of literacy levels on knowledge and attitudes towards preventive strategies against COVID-19, prospecting for the determinants, to allow for the design of specific and effective interventions of health education.

## 2. Materials and Methods

We conducted an observational cross-sectional study involving all students from the University of Porto, applying an online questionnaire. Although completely anonymous, students had to authenticate before answering to ensure that they answered the questionnaire only once.

The University of Porto is one of the biggest universities in Portugal. It has 14 faculties with approximately 20,000 students in 34 bachelor’s degrees and 18 integrated master’s degrees and more than 10,000 students in postgraduate education. All students enrolled at the University of Porto in the academic year 2021 were considered eligible for participation.

Data collection occurred from January 2021 to February 2021. All eligible students were invited to participate in the survey, on different occasions, through the institutional email system.

The self-answer questionnaire consisted of 3 parts.

In the first part, we measured health literacy through the Newest Vital Sign (NVS) questionnaire developed by Barry Weiss et al. [25], translated and validated to Portuguese by Anabela Martins et al. [26]. This instrument is a nutrition label compounded by six questions that require three minutes for administration. It has an internal consistency (Cronbach’s alpha) between 0.67 and 0.83. Using factor analysis with varimax rotation, two subscales explained 60.97% of the variance. Depending on the number of correct answers, we can classify the level of health literacy: a score of 0–1 suggests a high probability (50% or more) of limited literacy, a score of 2–3 indicates the possibility of limited literacy, and a score of 4–6 almost always indicates adequate literacy.

In the second section, we assessed students’ knowledge and attitudes towards preventive strategies related to COVID-19. For the elaboration of the questions, we used the guidelines for the implementation of non-pharmaceutical interventions against COVID-19 of the European Centre for Disease and Prevention Control [27].

Knowledge was assessed through 6 sentences: keeping physical distancing of 1–2 m, wearing shields/visors to replace masks, wearing gloves, wearing masks, hand hygiene, and avoiding contact of hands with the respiratory tract. We used a Likert scale of 5 points to assess students’ confidence in each preventive measure (from no trust to full confidence), dichotomized as correct (4–5) or not correct (1–3). Correct answers worth 1 point and wrong answers worth 0 points. The total score consisted of the sum of the punctuation. To be considered adequate knowledge, students should answer at least 5 questions correctly. We also evaluated participants’ knowledge about the symptoms related to COVID-19.

Attitudes were assessed through 7 sentences: keeping physical distancing of 1–2 m, wearing gloves, wearing masks indoors, wearing masks outdoors, hand hygiene, avoiding contact of hands with the respiratory tract, and respiratory hygiene. We also used a 5-point Likert scale (from never to always), dichotomized into correct (4–5) and not correct (1–3), with a total score varying from 0 to 7. Adequate attitudes corresponded to at least 6 correct answers.

The last part evaluated social and demographic variables: gender, age, education or practice in health, height, weight, nationality, civil status, family situation, the existence of children, the highest education level completed, current working condition, difficulty paying bills at the end of the month, easiness in buying medicines, easiness to access physician, and level in society (self-perception).

The sample size was established in a minimum of 765 participants, considering a margin of error of 3.5% for a confidence interval of 95%, without knowing the expectant results.

We used descriptive and inferential statistics. We used the logistic regression analysis, adjusted for gender, age, and the formation or practice in health, to assess the relationship between the literacy (total NVS score) and the adequacy of knowledge and attitudes. We also used a logistic regression multivariate model, adjusted for the same variables, to estimate the weight of each determinant in knowledge and attitudes about COVID-19. The relationship between the scores of knowledge and attitudes was assessed by a bivariate correlation of Pearson. The significance level was set at 0.05. Data were encoded and registered in a Microsoft Office Excel 2013^®^ database and analyzed using IBM SPSS Statistics^®^, version 27.0 (IBM Corp., Armonk, NY, USA).

The study protocol was assessed and approved by the Ethical Committee of Hospital de São João/Faculty of Medicine of University of Porto. In this investigation, we followed the principles of the Helsinki Declaration and the Oviedo Convention about the protection of human rights in biomedical investigation. The first page of the web form, before the questionnaire itself, included information for participants and asked for their explicit written consent, allowing for their refusal, which led to them automatically dropping out of the study.

## 3. Results

### 3.1. Participants’ Characteristics

A total of 871 students participated in the survey, all considered valid. As expected by the demography of students from the University of Porto, females represented the main proportion (76.3%), and the mean age was 23.9 (±7.0) years old (median = 22). Most of them were Portuguese (94.9%). The group with specific education or work practice in health represented 29.0% (*n* = 253). About a quarter presented at any time symptoms of COVID-19 (25.8%), and 83 (9.5%) referred to a confirmed infection. A total of 270 students (31.0%) had been in prophylactic isolation. Almost half of the participants had taken a diagnostic test (428; 49.1%). Table 1 shows the general demographic characteristics of our sample.

Literacy levels assessed by the NVS showed an adequate literacy in 801 participants (92%; 95%CI: 90.2%–93.8%), a possibility of limited literacy in 65 (7.5%; 95%CI: 5.7%–9.2%), and a high likelihood of limited literacy in 5 (0.6%; 95%CI: 0.1%–1.1%). There were no differences between gender or age.

### 3.2. Participants’ Knowledge and Attitudes towards Preventive Strategies Related to COVID-19

Table 2 shows the frequency of correct answers for each knowledge and attitudes question. Hand hygiene was the topic most hit in the knowledge evaluation by 99.3% of participants (95%CI: 98.8–99.9), and the indoor use of mask the most referred in attitudes by 98.9% of respondents (95%CI: 98.2–99.6). On the other hand, the use of gloves revealed some lack of knowledge at only 44.5% (95%CI: 41.2–47.8), as the attitudes towards the eviction of the contact of hands with respiratory tract was 81.2% (95%CI: 78.6–83.8). There was a positive, although weak, correlation between knowledge and attitudes (Pearson ρ = 0.180; *p* < 0.01).

Knowledge and attitudes scores were dichotomized into adequate or non-adequate. A total of 685 participants (78.6%; 95%CI: 77.2–80.0) presented adequate knowledge, and 787 participants (90.4%; 95%CI: 89.4–91.4) presented adequate attitudes.

### 3.3. Participants’ Knowledge about Symptoms

Concerning participants’ knowledge about the symptoms, they identified cough (86.7%), fever (84.0%), and fatigue (82.2%) as frequent or very frequent and vomiting (53.4%), nausea (48.1%), and diarrhea (43.6%) as less frequent or not frequent. Figure 1 shows the distribution of the answers.

### 3.4. Logistic Regression

Literacy is significantly associated with attitudes towards preventive strategies against COVID-19 (OR = 1.212; 95%CI: 1.002–1.467; *p* = 0.048), but not to knowledge (OR = 1.141; 95%CI: 0.981–1.326; *p* = 0.086), in the logistic regression model adjusted to age, gender, and education or practice on health. Gender and previous education significantly influenced the model (Table 3).

Other determinants of interest are in Figure 2. The analysis adjusted for gender, age, and health topics experience showed that having been infected (OR = 0.476; 95%CI: 0.289–0.784; *p* = 0.004) and having difficulty paying bills (OR = 0.531; 95%CI: 0.342–0.823; *p* = 0.005) were related to worse knowledge. It also showed that higher knowledge was associated with Portuguese nationality (OR = 3.143; 95%CI: 1.582–6.242; *p* = 0.001), high education level (OR = 1.305; 95%CI: 1.067–1.597; *p* = 0.01), easiness to access physician (OR: 1.507; 95%CI: 1.023–2.219; *p* = 0.038), and certain information sources, such as specialized books (OR = 1.556; 95%CI: 1.055–2.295; *p* = 0.026) and healthcare professionals (OR = 1.840; 95%CI: 1.225–2.766; *p* = 0.003). On the other hand, worse attitudes were apparent in students who were previously infected (OR = 0.502; 95%CI: 0.263–0.961; *p* = 0.037), and they were better in those who use specialized books (OR = 2.513; 95%CI: 1.436–4.398; *p* = 0.001) and healthcare providers (OR = 5.149; 95%CI: 1.734–15.288; *p* = 0.003) as their main sources of information.

## 4. Discussion

Literacy influences the behavior about COVID-19 protection. Our study shows a tendency for better knowledge to appear in those who have better literacy and a significant association between health literacy and attitudes. This is in concordance with a recent study in the USA population that found that adults with low coronavirus-related eHealth literacy are at a higher risk of having less protective COVID-19-related knowledge, attitudes, and practices [28]. During the pandemic, an enormous amount of information has been running continuously through the media, internet, social networks, and many other sources. Most of the regulatory agencies adopted a consistent speech focusing on the preventive measures against COVID-19, constantly repeated by many opinion makers, based on the recommendations published by the World Health Organization. Thus, even without thinking about it, people absorb the contents of these media messages and repeat them almost automatically. However, the relation is more complex when we ask for the attitudes as the way they put this information into action. Here, better literacy is associated with better attitudes, meaning that these individuals are better prepared to assume a change of behavior according to the available information. On the other hand, better knowledge is associated with better attitudes, but with a weak correlation, showing that information is crucial but not sufficient for changing the behaviors and activating them in a healthy life. People’s attitudes do not depend only on their knowledge, but also on their values, beliefs, emotions, convictions, and even social contexts. In literature, knowledge is described as one of the predictors of intention to adopt health promotion behaviors [29]. As we see, knowing the preventive strategies against COVID-19 does not mean that these will necessarily be reflected in attitudes. Previous studies document that health literacy is associated with personal preventive behaviors [30,31,32,33] and that limited health literacy is associated with lower adoption of protective behaviors, such as vaccinations, hand hygiene, and other self-care measures [34]. Another recent study in the USA population found that adults with low coronavirus-related eHealth literacy are at a higher risk of having less protective COVID-19-related knowledge, attitudes, and practices.

Additionally, we identified population characteristics that influenced knowledge and attitudes. The female gender relates to better knowledge and attitudes, which is consistent with previous studies [31,33]. Having education or practice on health also associates with better knowledge and attitudes, as expected, since these students are supposed to be prepared for health issues, with critical perception of health information, better knowledge about the diseases, medical skills, and prevention measures. Previous studies have shown that individuals with lower incomes and lower educational levels reported limited knowledge about health risks [31,35,36]. This may justify why our students, who have difficulty paying bills and have a low level of education, have worse knowledge. A previous infection is associated with worst knowledge and attitudes. This aspect could be explained by the fact that these people might be less careful, have less access to information, and consequently are more likely to become infected. Another way to look at this point is the asymptomatic character of most patients, especially in the younger population, leading to a depreciation of the disease itself after being infected.

Nevertheless, the evaluation of knowledge and attitudes towards preventive strategies against COVID-19 shows high values in both, although slightly higher in attitudes, perhaps partly due to the legal obligation of some of the preventive measures. During the period in which the answers were collected, wearing masks indoors and outdoors was mandatory by Portuguese law. According to our results, 98.9% and 94.9% of our participants wore masks indoors and outdoors, respectively. These findings are in line with previous studies that also document a high percentage of mask use [4,31]. We can assume the legal obligation as one reason for its use by some of our participants.

Our students identified cough, fever, and fatigue as the most frequent symptoms and nausea, vomiting, and diarrhea as the less frequent, which agrees with what we found in recent studies concerning the prevalence of symptoms [37,38,39,40,41]. The amount of information available and its dissemination might justify this correct identification of symptoms. One of the relevant measures to reduce the transmission of the disease is the fast and correct identification of cases, where the recognition of the symptoms is crucial.

Our study shows a high prevalence of adequate health literacy in 92% of participants (95%CI: 90.2%–93.8%) and a low prevalence of a high likelihood of limited health literacy in 0.6% of participants (95%CI: 0.1%–1.1%). These results are different from those found in previous research. In the Portuguese adaptation and validation of the Newest Vital Sign, Martins et al. (2014) found adequate health literacy in 76% of participants and a likelihood of limited health literacy in 10% [26]. Paiva et al. (2017) observed 27.1% (95%CI: 26.6–30.6) of their sample with adequate literacy and 42.5% (95%CI: 38.3–46.6) of their sample with a high likelihood of limited health literacy [42]. The differences found in our study can be explained by the composition of our sample. According to the literature, people with a complete college education are less likely to have limited health literacy compared to people with a less than 4th grade education [42]. This might explain the differences found as our population only includes university students. We found no significant differences between gender in line with previous studies [26,42,43,44]. Despite the common description of the increase of limited literacy with age [42,43,45], in our sample, the range of ages was small, explaining why we did not find it. On the other hand, limited literacy in older people is commonly associated with a lesser education, which was also not the case in our college students. Our findings are concordant with the European Health Literacy Survey [43], conducted in eight European countries. This survey found some specific subgroups where the proportion of people with limited health literacy was higher, such as financial deprivation, low social status, low education, or old age. When developing public health strategies to improve health literacy, these aspects must be considered. Students’ behaviors should be promoted by adequate government action and policies not only by providing health information, but also by providing health, social, and economic services for students to cope with the situation. We should develop policies and programs to encourage healthy and protective behavior, and adherence to COVID-19 prevention strategies [46]. It is also relevant to adjust curricula and train health professionals to better deal with health literacy challenges. The photo-novels (visual stories) are a new instrument available that can be used to provide vital information, resources, and support to vulnerable populations. In a recent study, they were used to improve health literacy by including a visual story within the text messages that showed people adopting healthy behaviors to protect against the virus [47].

Although there are already some studies that evaluate the relationship between digital health literacy and COVID-19 behaviors, as far as our knowledge, this is the first study measuring health literacy in college students, and one of the first in the world characterizing the impact of literacy on knowledge and attitudes towards preventive strategies against COVID-19. Nevertheless, we should interpret our results considering the limitations of this study. First, the cross-sectional design lacks temporality, opposing a longitudinal design that would have evaluated temporal relationships. Second, the sample was restricted to active students of the University of Porto, students who are all part of a highly educated population. This social context might have influenced the results. Furthermore, the questionnaire was sent through dynamic emails, and we are unable to guarantee that all emails were updated or that all students had access to their institutional email. Participation in the study was voluntary, which may have led to a selection bias, where students who participated may present different characteristics to those who did not participate. Additionally, the self-response questionnaire may lead to an information bias, in which students may respond according to what they believe to be socially acceptable instead of their own beliefs, despite the use of a Likert scale to prevent the tendency for the correct answer. Another aspect that can be considered a limitation of the study was the questionnaire that assessed knowledge and attitudes towards COVID-19. Although we used the ECDC guidelines for elaborating the questions, they were not part of a validated questionnaire. Even though it is not possible to extrapolate our results to non-college students, we believe that similar patterns may be present in other populations.

## 5. Conclusions

In conclusion, our study shows that better health literacy associates with better attitudes towards preventive strategies against COVID-19. Even without a significant impact on knowledge, these findings help to improve the global awareness among citizens and decision-makers towards health literacy as a tool to prevent communicable diseases. The results corroborate the expression that refers to literacy as the key for effective preventive medicine, because literacy empowers university students and all other population groups to take greater control in the prevention of COVID-19 and its spread, thus leading to better health outcomes. From a community and public health perspective, we should highlight programs that aim to improve health literacy. Possible strategies include improving the quality of available health information about COVID-19 and training to improve general or coronavirus-specific search skills. Adequate government action and policies should provide health information as well as social and economic services, making literacy for health a fundamental skill in college students, who are the future professionals in different subject areas.

## Figures and Tables

**Figure 1 ijerph-18-05421-f001:**
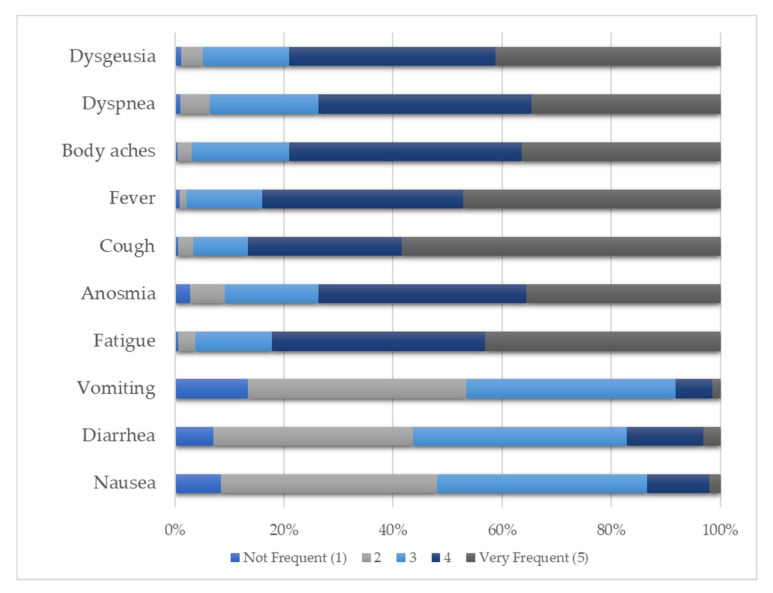
Participants’ knowledge about the symptoms related to COVID-19.

**Figure 2 ijerph-18-05421-f002:**
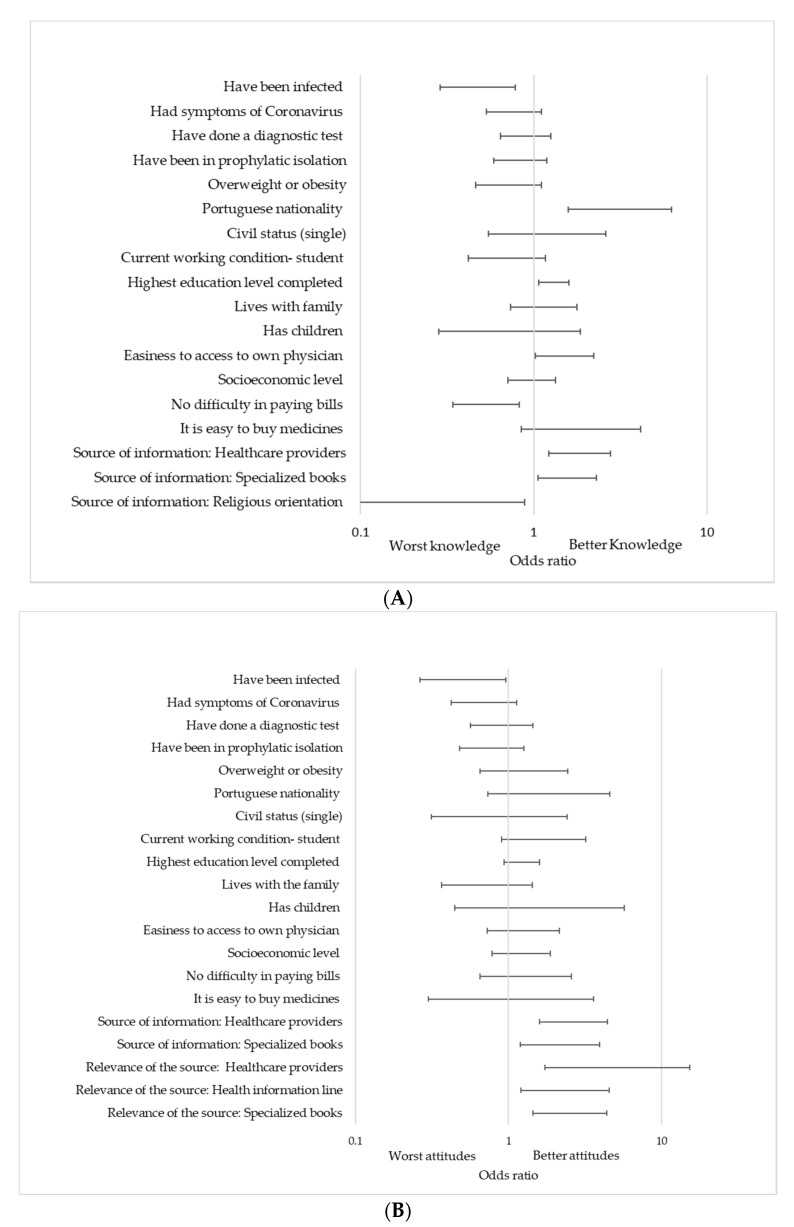
(**A**) Knowledge and (**B**) attitudes about COVID-19.

**Table 1 ijerph-18-05421-t001:** Sociodemographic characteristics.

Characteristics	*n* = 871 (%)
Gender	
Female	665 (76.3%)
Male	206 (23.7%)
Mean age (SD)	23.9 (±7.0)
Practice or education in health	253 (29.0%)
Overweight/Obesity	111 (12.7%)/28 (3.2%)
Had symptoms of Coronavirus	225 (25.8%)
Have been infected	83 (9.5%)
Have been in prophylactic isolation	270 (31.0%)
Have taken a diagnostic test	428 (49.1%)
Nationality	
Portuguese	827 (94.9%)
Other	43 (4.9%)
Do not know/did not answer	1 (0.1%)
Civil status	
Single	781 (89.7%)
Married	78 (9.0%)
Other	7 (0.8%)
Do not know/did not answer	5 (0.6%)
Family situation	
Lives alone	28 (3.2%)
Lives with the household	717 (82.3%)
Lives away from home	119 (13.7%)
Do not know/did not answer	7 (0.8%)
Highest education level completed	
Upper secondary education	440 (50.5%)
Bachelor	205 (23.5%)
Post-graduation	221 (25.3%)
Do not know/did not answer	5 (0.6%)
Current working condition	
Student	663 (76.1%)
Student workers (paid or unpaid)	180 (20.7%)
Other	25 (2.8%)
Do not know/did not answer	3 (0.3%)
Difficulty in paying bills at the end of the month (during the past 12 months)	
Most times and sometimes	136 (15.6%)
Almost never and never	619 (71.1%)
Do not know/did not answer	116 (13.3%)
Easiness in buying medicines	
Very easy and easy	772 (88.6%)
Difficult and very difficult	33 (3.7%)
Do not know/did not answer	66 (7.6%)
Easiness to access to own physician	
Very easy and easy	564 (64.8%)
Difficult and very difficult	216 (24.8%)
Do not know/did not answer	91 (10.4%)
Socioeconomic level (self-perception)	
Low	25 (2.9%)
Middle	488 (56%)
High	332 (38.1%)
Did not answer	26 (3%)

**Table 2 ijerph-18-05421-t002:** (**A**) Frequency of adequate responses to each question that assessed knowledge. (**B**) Frequency of adequate responses to each question that assessed attitudes.

	%	95%CI
**(A) Knowledge**		
The minimum distance required to ensure a safe contact is 2 m.	89.4	87.4–91.4
The use of a visor replaces the use of a mask.	81.2	78.6–83.8
There is a proven benefit in the use of gloves.	44.5	41.2–47.8
There is a proven benefit in wearing a mask.	97.5	96.5–98.5
Hand washing/disinfection should be a frequent practice in everyday life.	99.3	98.8–99.9
We must avoid contact of the hands with the respiratory tract.	97.9	96.9–98.9
**(B) Attitudes**		
Do you maintain the recommended social distance?	91.6	89.8–93.4
Do you wear gloves?	89.8	87.8–91.8
Do you wear a mask indoors?	98.9	98.2–99.6
Do you wear a mask outdoors?	94.9	93.4–96.4
Do you wash your hands?	97.2	96.1–98.3
Do you avoid contact of the hands with the respiratory tract?	81.2	78.6–83.8
Do you adopt respiratory hygiene measures?	94.7	93.2–96.2

**Table 3 ijerph-18-05421-t003:** Relation between (**A**) knowledge or (**B**) attitudes towards COVID-19 and literacy for health, adjusted for age, gender, and practice or formation in health.

	(A) Knowledge			(B) Attitudes		
	OR	95%CI	*p*	OR	95%CI	*p*
Total NVS Score	1.141	0.981–1.326	0.086	1.212	1.002–1.467	0.048
Gender	0.579	0.401–0.835	0.003	0.541	0.333–0.879	0.013
Practice or formation in health	1.705	1.134–2.562	0.010	3.974	1.873–8.429	<0.001
Age	1.009	0.984–1.034	0.492	0.991	0.962–1.022	0.573

## Data Availability

The data presented in this study are available on request from the corresponding author. The data are not publicly available due to ethical and privacy constraints.

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
