# Peer review of "The Impact of Health Literacy on Knowledge and Attitudes towards Preventive Strategies against COVID-19: A Cross-Sectional Study"

_ijerph, 2021, doi:10.3390/ijerph18105421_

Round 1
Reviewer 1 Report
The article is interesting to read, contains original data analysis, and adds to our understanding.
The article is comprehensive and structured in a way that allows the reader to follow it easily. But there are a few areas in that need more previous results/ clarification.
Introduction: Health literacy --- for example Nutbeam & Lloyd 2021
Covid 19 --- for example Gautam, Dileepan et a. 2021, Montagniet et al. 2021, Dadaczynski et al. 2021, Dodd et al. 2021
Discussion: there are too many short chapters, for example line 256-259, 260-265, 266-269, 270-274 ect. This is a recommendation to help align the reader with the author(s)' intent.
Conclusions: Conclusions is too abstract level, please write some concrete ideas
Author Response
The article is interesting to read, contains original data analysis, and adds to our understanding.
The article is comprehensive and structured in a way that allows the reader to follow it easily. But there are a few areas in that need more previous results/ clarification.
Introduction: Health literacy --- for example Nutbeam & Lloyd 2021
Covid 19 --- for example Gautam, Dileepan et a. 2021, Montagniet et al. 2021, Dadaczynski et al. 2021, Dodd et al. 2021”
Thank you for the literature suggestions. We reviewed the introduction accordingly have added a section reviewing previous literature Lines 53-54, 83-86 and 91-98.
“Discussion: there are too many short chapters, for example line 256-259, 260-265, 266-269, 270-274 ect. This is a recommendation to help align the reader with the author(s)' intent.”
We have changed the discussion section. (line 252 and following)
“Conclusions: Conclusions is too abstract level, please write some concrete ideas.”
We have completed the conclusion section by adding concrete ideas. Lines 379-419
Reviewer 2 Report
The study titled "The Impact of Health Literacy on Knowledge and Attitudes to-2 wards preventive strategies against COVID-19: a cross-sectional 3 study" is an assessment of health literacy, knowledge and attitude towards COVID-19.
Strengths
- Well laid out experimental plan
- Pitfalls adequately discussed.
Weakness
- Limited cohort of university students.
- Line 270-271 may be better rephrased.
Author Response
“The study titled "The Impact of Health Literacy on Knowledge and Attitudes to-2 wards preventive strategies against COVID-19: a cross-sectional 3 study" is an assessment of health literacy, knowledge and attitude towards COVID-19.
Strengths
- Well laid out experimental plan
- Pitfalls adequately discussed.
Weakness
- Limited cohort of university students.”
Thank you for the comment. As discussed in lines 358 and following, there are several constraints due to the methodological options we made. Nevertheless, we do not think they call into question our conclusions
- “Line 270-271 may be better rephrased.”
We have rephrased this line from “A previous infection is associated with worst knowledge and attitudes. This aspect could be explained by the fact that these people might be careless, have less information, and consequently more likely to become infected” to “This aspect could be explained by the fact that these people might be less careful, have less access to information, and consequently more likely to become infected.”
Reviewer 3 Report
The paper is a very interesting one and has a lot of potentials. Still, there are some opportunity areas to be attended, in order to enhance its quality and impact.
I have the next mandatory suggestions to be made:
- The abstract is too technical. It appears more a sort of Statistical summary than an abstract. please write by saying what are you doing?, Why are you doing it? how are you doing it? what are your results and implication to knowledge in the subject? and What are the main contributions in your paper?
- The paper lacks a previous literature review. It is important and mandatory to have this section in order to justify the methods used in the paper. Also, the review must be made to justify your motivations in the paper and to discuss what are the results, conclusions, and gaps to be filled with your world. This section is mandatory.
- Your discussions must be compared to similar studies that are in other places (regions, countries, etc.) and to similar studies in other study objects (other epidemic episodes or similar).
- Table 1 is too big. Please use charts (pie or bar charts) to summarize more your results. Also, the inclusion of parentheses with the 95% confidence interval leads to a fussy reading. please present only the values and make clearer your results exposition.
- With all due respect, I can't see a proper results discussion. I read an unarticulated collection of statistical data of your results. please enhance the writing style and discussion.
- Also, please justify why are you presenting the students of the University of Porto. What is the impact of your results on the health community in Portugal but, most of all, on the rest of the world? Justify why is your study object so important to the scope of the journal.
- Please, Also discuss and justify properly why is this important for the Academic and Public (and private) health institutions community who is the target of the journal? Give this answer in light of your results and the previous works (literature) review.
- Do you have a hypothesis to be tested?
- The statements made in lines 241 to 244 must be proven. This in light of previous works that proved them as true.
- I don't see necessary the Discussion section if you make a proper results review and discussion in the previous section. You must merge both sections to have a more straightforward review. In the current form, the results collection and its discussion are not properly articulated with a proper rhetorical situation.
- Does the opinion or view of Porto University's students reflect the view and thinking of the rest of the Portuguese or even European people? Please explain and support your comments. If there is no link between this sample results and the rest of Portugal's population, the paper could be weaker than it seems (as I said before, it has potential).
- The limitations of your work and guidelines for further research must be presented in the Conclusions section. Please correct and enhance.
- Following this last suggestion, please enhance the conclusions section. It is too small and weak. The paper (or the potential idea) is not small and weak.
Author Response
Reviewer 3:
“The paper is a very interesting one and has a lot of potentials. Still, there are some opportunity areas to be attended, in order to enhance its quality and impact.
I have the next mandatory suggestions to be made:
1)The abstract is too technical. It appears more a sort of Statistical summary than an abstract. please write by saying what are you doing?, Why are you doing it? how are you doing it? what are your results and implication to knowledge in the subject? and What are the main contributions in your paper?”
We changed the abstract accordingly
“2) The paper lacks a previous literature review. It is important and mandatory to have this section in order to justify the methods used in the paper. Also, the review must be made to justify your motivations in the paper and to discuss what are the results, conclusions, and gaps to be filled with your world. This section is mandatory.”
We reviewed the introduction accordingly have added a section reviewing previous literature Lines 53-54, 83-86 and 91-98.
“3) Your discussions must be compared to similar studies that are in other places (regions, countries, etc.) and to similar studies in other study objects (other epidemic episodes or similar).”
We added comparisons of our results with previous studies in the discussion session- line 253-256. We have already part of our results compared with similar studies- line 271-274; 278-279; 282-284; 296-298; 299-301; 307-311; 315-318 and 320-321.
“4) Table 1 is too big. Please use charts (pie or bar charts) to summarize more your results. Also, the inclusion of parentheses with the 95% confidence interval leads to a fussy reading. please present only the values and make clearer your results exposition.”
We simplified the table 1 and put some of the information in the text.
“5) With all due respect, I can't see a proper results discussion. I read an unarticulated collection of statistical data of your results. please enhance the writing style and discussion.”
Discussion is provided in the section “Discussion”
“6) Also, please justify why are you presenting the students of the University of Porto. What is the impact of your results on the health community in Portugal but, most of all, on the rest of the world? Justify why is your study object so important to the scope of the journal.” 7) Please, Also discuss and justify properly why is this important for the Academic and Public (and private) health institutions community who is the target of the journal? Give this answer in light of your results and the previous works (literature) review.”
Many times literacy analysis is conditioned by the education status, and the conclusion is that people present low literacy and consequently low knowledge and attitudes because of the accordingly low education. Using college students accustomed to scientific research and internet utilization allows us to overcome this problem and to measure the impact of literacy for health independently of educational constraints. University of Porto is one of the biggest colleges in Portugal, and we believe it is representative of the remaining Portuguese students and not far from the European situation. University students are the future professionals in their areas. An healthy campus will certainly reflect on higher healthy lives.
“8) Do you have a hypothesis to be tested?”
Yes, of course. The main hypothesis is contextualized in the objective of our research. It tests the relationship between the status of literacy (independent variable) and the knowledge and attitudes towards the covid-19 (dependent variable).
“9) The statements made in lines 241 to 244 must be proven. This in light of previous works that proved them as true.”
Corrected. The recommendations were based on the WHO statements.
“10) I don't see necessary the Discussion section if you make a proper results review and discussion in the previous section. You must merge both sections to have a more straightforward review. In the current form, the results collection and its discussion are not properly articulated with a proper rhetorical situation.”
Thank you for the comment. The option of the headings is conforming to the instructions for authors, in the manuscript preparation section.
“11) Does the opinion or view of Porto University's students reflect the view and thinking of the rest of the Portuguese or even European people? Please explain and support your comments. If there is no link between this sample results and the rest of Portugal's population, the paper could be weaker than it seems (as I said before, it has potential).”
Yes, as stated earlier (please, see our answer to comment 6)
“12) The limitations of your work and guidelines for further research must be presented in the Conclusions section. Please correct and enhance. “
According the strobe statement for cross-sectional studies, the limitations shall be discussed on the discussion section, but we are available to change it if necessary.
The proposal for further research was added in the conclusion section.
“13) Following this last suggestion, please enhance the conclusions section. It is too small and weak. The paper (or the potential idea) is not small and weak.”
Done
Round 2
Reviewer 3 Report
I'm happy with all the changes except one:
Please present a clear section 2: a literature review. Maybe I didn't explain myself (an apology). It is important to have this section for 2 reasons:
- There, you will discuss all the previous works and suggest what must be done and How are you extending the current works. This section is important for unrelated readers or readers of other disciplines that will read and cite your paper. That is why this section Must be in the paper.
- Also in that section, you will express clearly your hypothesis and not imply it in the text e.g. "Given the previous literature our hypothesis to be tested is: 2+2 is not always 4" or "Given the literature review and background given in the introduction section our goal (or 'our position') is to prove that 2 +2 is not always 4".
In order to be accepted, the paper must incorporate that improvement. An express literature review section (with a discussion of it) and your hypothesis in that section will enhance the good quality of your paper.
Author Response
Please present a clear section 2: a literature review. Maybe I didn't explain myself (an apology). It is important to have this section for 2 reasons:
- There, you will discuss all the previous works and suggest what must be done and How are you extending the current works. This section is important for unrelated readers or readers of other disciplines that will read and cite your paper. That is why this section Must be in the paper.
- Also in that section, you will express clearly your hypothesis and not imply it in the text e.g. "Given the previous literature our hypothesis to be tested is: 2+2 is not always 4" or "Given the literature review and background given in the introduction section our goal (or 'our position') is to prove that 2 +2 is not always 4".
In order to be accepted, the paper must incorporate that improvement. An express literature review section (with a discussion of it) and your hypothesis in that section will enhance the good quality of your paper.
Thank you for the comments.
We agree with the reviewer that a good literature review is crucial for the organization of the research and of course for the article, too.
The main sections of the manuscript are settled in the instructions for authors: introduction, Materials & Methods, Results, Discussion and Conclusions. This is concordant with the most often used format of scientific publication for original research.
Although maintaining this structure, we reviewed the introduction to make more clear the literature review and the rational for the research question.
We also reviewed the objectives/hypothesis to make them more clear for the readers.
Following the recommendation of guideline STROBE for cross-sectional studies, the most of the literature review is placed in the discussion section.